# Understanding intermediate layers using linear classifier probes

**Guillaume Alain & Yoshua Bengio**
Department of Computer Science and Operations Research
Université de Montréal
Montreal, QC. H3C 3J7
`guillaume.alain.umontreal@gmail.com`

## Abstract

Neural network models have a reputation for being black boxes. We propose a new method to better understand the roles and dynamics of the intermediate layers. This has direct consequences on the design of such models and it enables the expert to be able to justify certain heuristics (such as adding auxiliary losses in middle layers). Our method uses linear classifiers, referred to as "probes", where a probe can only use the hidden units of a given intermediate layer as discriminating features. Moreover, these probes cannot affect the training phase of a model, and they are generally added after training. They allow the user to visualize the state of the model at multiple steps of training. We demonstrate how this can be used to develop a better intuition about models and to diagnose potential problems.

## 1 Introduction

The recent history of deep neural networks features an impressive number of new methods and technological improvements to allow the training of deeper and more powerful networks.

Despite this, models still have a reputation for being black boxes. Neural networks are criticized for their lack of interpretability, which is a tradeoff that we accept because of their amazing performance on many tasks. Efforts have been made to identify the role played by each layer, but it can be hard to find a meaning to individual layers.

There are good arguments to support the claim that the first layers of a convolution network for image recognition contain filters that are relatively "general", in the sense that they would work great even if we switched to an entirely different dataset of images. The last layers are specific to the dataset being used, and have to be retrained when using a different dataset. In Yosinski et al. (2014) the authors try to pinpoint the layer at which this transition occurs, but they show that the exact transition is spread across multiple layers.

In this paper, we introduce the concept of *linear classifier probe*, referred to as a "probe" for short when the context is clear. We start from the concept of *Shannon entropy*, which is the classic way to describe the information contents of a random variable. We then seek to apply that concept to understand the roles of the intermediate layers of a neural network, to measure how much information is gained at every layer (answer : technically, none). We argue that it fails to apply, and so we propose an alternative framework to ask the same question again. This time around, we ask what would be the performance of an optimal linear classifier if it was trained on the inputs of a given layer from our model. We demonstrate how this powerful concept can be very useful to understand the dynamics involved in a deep neural network during training and after.

## 2 Information theory

It was a great discovery when Claude Shannon repurposed the notion of *entropy* to represent information contents in a formal way. It laid the foundations for the discipline of information theory. We would refer the reader to first chapters of MacKay (2003) for a good exposition on the matter.

Naturally, we would like to ask some questions about the information contents of the many layers of convolutional neural networks.

- What happens when we add more layers?
- Where does information flow in a neural network with multiple branches?
- Does having multiple auxiliary losses help? (e.g. Inception model)

Intuitively, for a training sample $x_i$ with its associated label $y_i$, a deep model is getting closer to the correct answer in the higher layers. It starts with the difficult job of classifying $x_i$, which becomes easier as the higher layers distill $x_i$ into a representation that is easier to classify. One might be tempted to say that this means that the higher layers have more *information* about the ground truth, but this would be incorrect.

Here there is a mismatch between two different concepts of information. The notion of entropy *fails* to capture the essence of those questions. This is illustrated in a formal way by the *Data Processing Inequality*. It states that, for a set of three random variables satisfying the dependency

$$X \rightarrow Y \rightarrow Z$$

then we have that

$$I(X; Z) \leq I(X; Y)$$

where $I(X, Y)$ is the mutual information.

Intuitively, this means that the deterministic transformations performed by the many layers of a deep neural network are not adding more information. In the best case, they preserve information and affect only the representation. But in almost all situations, they lose some information in the process.

If we distill this further, we can think of the serious mismatch between the two following ideas :

- Part of the genius of the notion of entropy is that is distills the essence of information to a quantity that does not depend on the particular representation.
- A deep neural network is a series of simple deterministic transformations that affect the representation so that the final layer can be fed to a linear classifier.

The former ignores the representation of data, while the latter is an expert in finding good representations. A deaf painter is working on a visual masterpiece to offer to a blind musician who plays music for him.

We need a conceptual tool to analyze neural networks in a way that corresponds better to our intuitive notion of information. The role of data representation is important, but we would also argue that we have to think about this issue as it relates to computational complexity. A linear classifier is basically the simplest form of classifier that is neither trivial nor degenerate.

We define a new notion of information that depends on our ability to classify features of a given layer with an optimal linear classifier. Then we have a conceptual tool to ask new questions and to get potentially interesting answers.

We end this section with a conceptual example in Figure 1. If $X$ contains an image of the savannah, and $Y \in \{0, 1\}$ refers to whether it contains a lion or not, then none of the subsequent layers are truly more informative than $X$ itself. The raw bits from the picture file contain everything.

## 3    LINEAR CLASSIFIER PROBES

In section 3.1 we present the main concept of this paper. We illustrate the concept in section 3.3. We then present a basic experiment in section 3.4. In section 3.6 we modify a very deep network in two different ways and we show how probes allow us to visualize the consequences (sometimes disastrous) of our design choices.

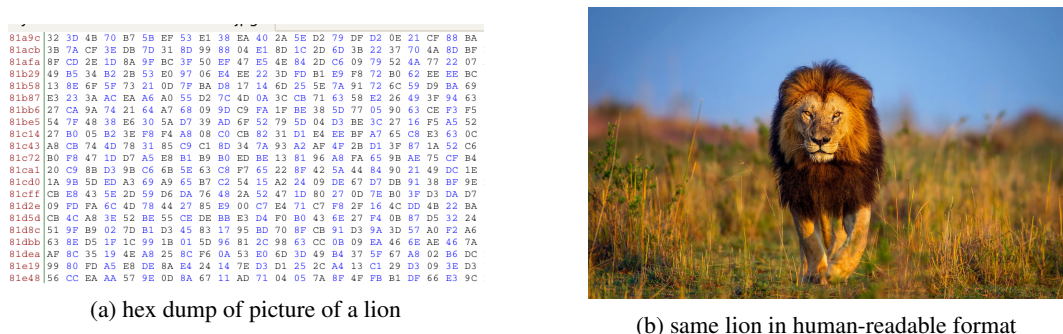

(a) hex dump of picture of a lion (b) same lion in human-readable format

Figure 1: The hex dump represented on the left has more information contents than the image on the right. Only one of them can be processed by the human brain in time to save their lives. Computational convenience matters. Not just entropy.

## 3.1 PROBES

As we discussed the previous section, there is indeed a good reason to use many deterministic layers, and it is because they perform useful transformations to the data with the goal of *ultimately fitting a linear classifier at the very end*. That is the purpose of the many layers. They are a tool to transform data into a form to be fed to a boring linear classifier.

With this in mind, it is natural to ask if that transformation is sudden or progressive, and whether the intermediate layers already have a representation that is immediately useful to a linear classifier. We refer the reader to Figure 2 for a diagram of probes being inserted in the usual deep neural network.

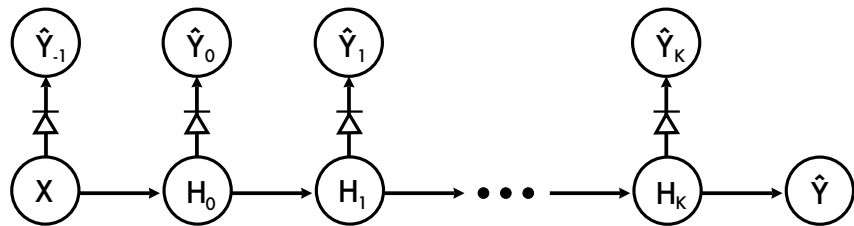

Figure 2: Probes being added to every layer of a model. These additional probes are not supposed to change the training of the model, so we add a little diode symbol through the arrows to indicate that the gradients will not backpropagate through those connections.

The conceptual framework that we propose is one where the intuitive notion of *information* is equivalent with *immediate suitability for a linear classifier* (instead of being related to entropy).

Just to be absolutely clear about what we call a *linear classifier*, we mean a function

$$f : H \to [0, 1]^D$$
$$h \mapsto \text{softmax}\,(Wh + b)\,.$$

where $h \in H$ are the features of some hidden layer, $[0, 1]^D$ is the space of one-hot encodings of the $D$ target classes, and $(W, b)$ are the probe weights and biases to be learned so as to minimize the usual cross-entropy loss.

Over the course of training a model, the parameters of the model change. However, probes only make sense when we refer to a given training step. We can talk about the probes at iteration $n$ of training, when the model parameters are $\theta_n$. **These parameters are not affected by the probes.** We prevent backpropagation through the model either by stopping the gradient flow (done with `tf.stop_gradient` in tensorflow), or simply by specifying that the only variables to be updated are the probe parameters, while we keep $\theta_n$ frozen.

### 3.1.1 TRAINING THE PROBES

For the purposes of this paper, we train the probes up to convergence with fixed model parameters, and we report the prediction error on the training set.

It is absolutely possible to train the probes simultenously while training the model itself. This is a good approach if we consider about how long it can take to train the model. However, this creates a potential problem if we optimize the loss of the model more quickly than the loss of the probes. This can present a skewed view of the actual situation that we would have if we trained the probes until convergence before updating the model parameters. If we accept this trade off, then we can train the probes at the same time as the model.

In some situations, the probes might overfit the training set, so we may want to do early stopping on the validation set and report the performance for the probes on the test set. This is what we do in section 3.4 with the simple MNIST convnet.

We are still unsure if one of those variations should be preferred in general, and right now they all seem acceptable so long as we interpret the probe measurements properly.

Note that training those probes represents a convex optimization problem. In practice, this does mean guarantee that they are easy to train. However, it is reassuring because it means that probes taken at time $\theta_n$ can be used as initialization for probes at time $\theta_{n+1}$.

We use cross-entropy as probe loss because all models studied here used cross-entropy. Other alternative losses could be justified in other settings.

### 3.2 PROBES ON BIFURCATING TOY MODEL

Here we show a hypothetical example in which a model contains a bifurcation with two paths that later recombine. We are interested in knowing whether those two branches are useful, or whether one is potentially redundant or useless.

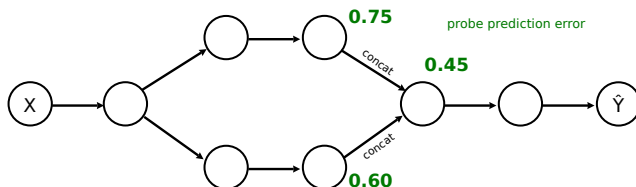

For example, the two different branches might contain convolutional layers with different dimensions. They may have a different number of sublayers, or one might represent a skip connection. We assume that the branches are combined through concatenation of their features, so that nothing is lost.

For this hypothetical situation, we indicate the probe prediction errors on the graphical model. The upper path has a prediction error of $0.75$, the lower path has $0.60$, and their combination has $0.45$. Small errors are preferred. Although the upper path has "less information" than the lower path, we can see here that it is not redundant information, because when we concatenate the features of the two branches we get a prediction error of $0.45 < 0.60$.

If the concatenated layer had a prediction error of $0.60$ instead of $0.45$, then we could declare that the above branch did nothing useful. It may have nonzero weights, but it's still useless.

Naturally, this kind of conclusion might be entirely wrong. It might be the case that the branch above contains very meaningful features, and they simply happen to be useless to a linear classifier applied right there. The idea of using linear classification probes to understand the roles of different branches is suggested as a heuristic instead of a hard rule. Moreover, if the probes are not optimized perfectly, the conclusions drawn can be misleading.

Note that we are reporting here the prediction errors, and it might be the case that the loss is indeed lower when we concatenate the two branches, but for some reason it could fail to apply to the prediction error.

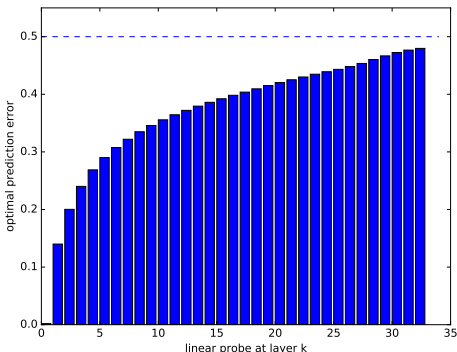

Figure 3: Toy experiment described in section 3.3, with linearly separable data (two labels), an untrained MLP with 32 layers, and probes at every layer. We report the prediction error for every probe, where 0.50 would be the performence of a coin flip and 0.00 would be ideal. Note that the layer 0 here corresponds to the raw data, and the probes are indeed able to classify it perfectly. As expected, performance degrades when applying random transformations. If many more layers were present, it would be hard to imagine how the final layer (with the model loss) can get any useful signal to backpropagate.

## 3.3 PROBES ON UNTRAINED MODEL

We start with a toy example to illustrate what kind of plots we expect from probes. We use a 32-layer MLP with 128 hidden units. All the layers are fully-connected and we use LeakyReLU(0.5) as activation function.

We will run the same experiment 100 times, with a different toy dataset each time. The goal is to use a data distribution $(X, Y)$ where $X \in \mathbb{R}^{128}$ is drawn $\mathcal{N}(0, I)$ and where $Y \in \{-1, 1\}$ in linearly separable (i.e. super easy to classify with a one-layer neural network). To do this, we just pick a $w \in \mathbb{R}^{128}$ for each experiment, and let the label $y_n$ be the sign of $x_n^T w$.

We initialize this 32-layer MLP using *glorot_normal* initialization, we do not perform any training on the model, and we add one probe at every layer. We optimize the probes with RMSProp and a sufficiently small learning rate.

In Figure 3, we show the prediction error rate for every probe, averaged over the 100 experiments. The graph includes a probe applied directly on the inputs $X$, where we naturally have an error rate that is essentially zero (to be expected by the way we constructed our data), and which serves as a kind of sanity check. Given that we have only two possible labels, we also show a dotted horizontal line at 0.50, which is essentially the prediction error that we would get by flipping a coin. We can see that the prediction error rate climbs up towards 0.50 as we go deeper in the MLP (with untrained parameters).

This illustrates the idea that the input signal is getting mangled by the successive layers, so much that it becomes rather useless by the time we reach the final layer. We checked the mean activation norm of the hidden units at layer 32 to be sure that numerical underflow was not the cause for the degradation. Note that this situation could be avoided by using orthogonal weights.

One of the popular explanation for training difficulties in very deep models is that of the exploding/vanishing (Hochreiter, 1991; Bengio et al., 1993). Here we would like to offer another complementary explanation, based on the observations from Figure 3. That is, at the beginning of training, the usefulness of layers decays as we go deeper, reaching the point where the deeper layers are utterly useless. The values contained in the last layer are then used in the final softmax classifier, and the loss backpropagates the values of the derivatives. Since that derivative is based on garbage activations, the backpropagated quantities are also garbage, which means that the weights are all going to be updated based on garbage. The weights stay bad, and we fail to train the model. The authors like to refer to that phenomenon as *garbage forwardprop, garbage backprop*, in reference to the popular concept of *garbage in, garbage out* in computer science.

## 3.4 PROBES ON MNIST CONVNET

In this section we run the MNIST convolutional model provided by the `tensorflow` github repo (`tensorflow/models/image/mnist/convolutional.py`) We selected that model for reproducibility and to demonstrate how to easily peek into popular models by using probes.

We start by sketching the model in Figure 4. We report the results at the beginning and the end of training on Figure 5. One of the interesting dynamics to be observed there is how useful the first layers are, despite the fact that the model is completely untrained. Random projections can be useful to classify data, and this has been studied by others (Jarrett et al., 2009).

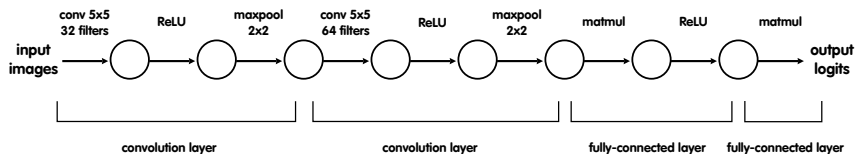

Figure 4: This graphical model represents the neural network that we are going to use for MNIST. The model could be written in a more compact form, but we represent it this way to expose all the locations where we are going to insert probes. The model itself is simply two convolutional layers followed by two fully-connected layer (one being the final classifier). However, we insert probes on each side of each convolution, activation function, and pooling function. This is a bit overzealous, but the small size of the model makes this relatively easy to do.

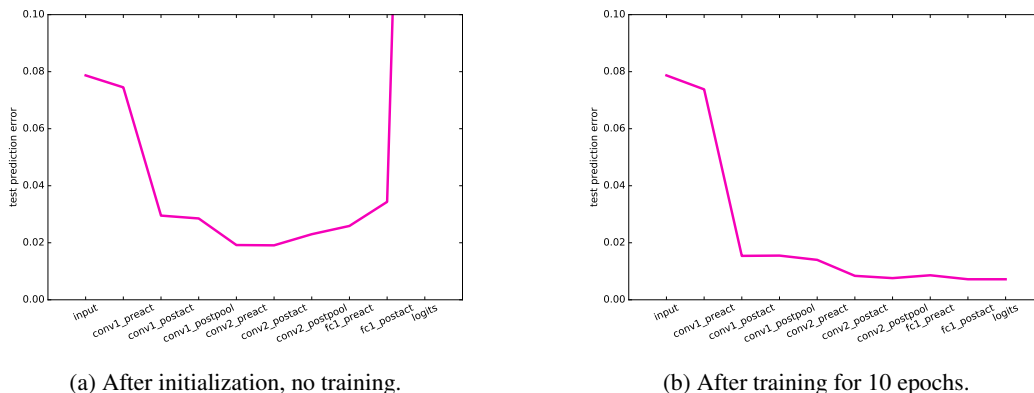

(a) After initialization, no training.    (b) After training for 10 epochs.

Figure 5: We represent here the test prediction error for each probe, at the beginning and at the end of training. This measurement was obtained through early stopping based on a validation set of $10^4$ elements. The probes are prevented from overfitting the training data. We can see that, at the beginning of training (on the left), the randomly-initialized layers were still providing useful trans-formations. The test prediction error goes from 8% to 2% simply using those random features. The biggest impact comes from the first ReLU. At the end of training (on the right), the test prediction error is improving at every layer (with the exception of a minor kink on `fc1_preact`).

## 3.5 PROBES ON INCEPTION V3

We have performed an experiment using the Inception v3 model on the ImageNet dataset (Szegedy et al., 2015; Russakovsky et al., 2015). This is very similar to what is presented in section 3.4, but on a much larger scale. Due to the challenge presented by this experiment, we were not able to do everything that we had hoped. We have chosen to put those results in the appendix section A.2.

Certain layers of the Inception v3 model have approximately one million features. With 1000 classes, this means that some probes can take even more storage space than the whole model it-self. In these cases, one of the creative solutions was to try to use only a random subset of the features. This is discussed in the appendix section A.1.

## 3.6 AUXILIARY LOSS BRANCHES AND SKIP CONNECTIONS

Here we investigate two ways to modify a deep model in order to facilitate training. Our goal is not to convince the reader that they should implement these suggestions in their own models. Rather, we want to demonstrate the usefulness of the linear classifier probes as a way to better understand what is happening in their deep networks.

In both cases we use a toy model with 128 fully-connected layers with 128 hidden units in each layer. We train on MNIST, and we use Glorot initialization along with leaky ReLUs.

We choose this model because we wanted a *pathologically deep* model without getting involved in architecture details. The model is pathological in the sense that smaller models can easily be designed to achieve better performance, but also in the sense that the model is so deep that it is very hard to train it with gradient descent methods. From our experiments, the maximal depth where things start to break down was depth 64, hence the choice here of using depth 128.

In the first scenario, we add one linear classifier at every 16 layers. These classifiers contribute to the loss minimization. They are not probes. This is very similar to what happens in the famous Inception model where "auxiliary heads" are used (Szegedy et al., 2015). This is illustrated in Figure 6a, and it works nicely. The untrainable model is now made trainable through a judicious use of auxiliary classifier losses. The results are shown in Figure 7.

In the second scenario, we look at adding a bridge (a skip connection) between layer 0 and layer 64. This means that the input features to layer 64 are obtained by concatenating the output of layer 63 with the features of layer 0. The idea here is that we might observe that the model would effectively train a submodel of depth 64, using the skip connection, and shift gears later to use the whole depth of 128 layers. This is illustrated in Figure 6b, and the results are shown in Figure 8. It does not work as expected, but the failure of this approach is visualized very nicely with probes and serves as a great example of their usefulness in diagnosing problems with models.

In both cases, there are two interesting observations that can be made with probes. We refer readers to `https://youtu.be/x8j4ZHCR2FI` for the full videos associated to Figures 5, 7 and 8.

Firstly, at the beginning of training, we can see how the raw data is directly useful to perform linear classification, and how this degrades as more layers are added. In the case of the skip connection in Figure 8, this has the effect of creating two bumps. This is because the layer 64 also has the input data as direct parent, so it can fit a probe to that signal.

Secondly, the prediction error goes down in all probes during training, but it does so in a way that starts with the parents before it spreads to their descendants. This is even more apparent on the full video (instead of the 3 frames provided here). This is a ripple effect, where the prediction error in Figure 6b is visually spreading like a wave from the left of the plot to the right.

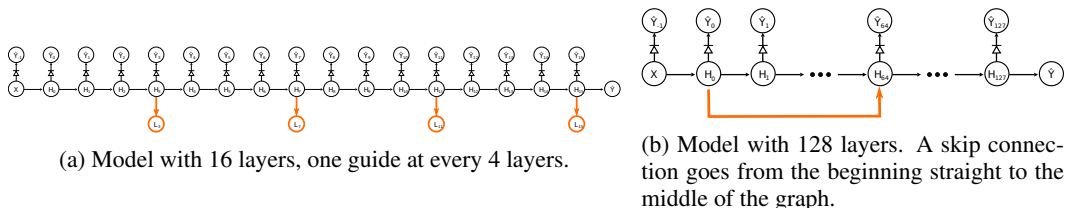

(a) Model with 16 layers, one guide at every 4 layers.

(b) Model with 128 layers. A skip connection goes from the beginning straight to the middle of the graph.

Figure 6: Examples of deep neural network with one probe at every layer (drawn above the graph). We show here the addition of extra components to help training (under the graph, in orange).

## 4    DISCUSSION AND FUTURE WORK

We have presented more toy models or simple models instead of larger models such as Inception v3. In the appendix section A.2 we show an experiment on Inception v3, which proved to be more challenging than expected. Future work in this domain would involve performing better experiments on a larger scale than small MNIST convnets, but still within a manageable size so we can properly train all the probes. This would allow us to produce nice videos showing many training steps in sequence.

We have received many comments from people who thought about using multi-layer probes. This can be seen as a natural extension of the linear classifier probes. One downside to this idea is that we lose the convexity property of the probes. It might be worth pursuing in a particular setting, but as of

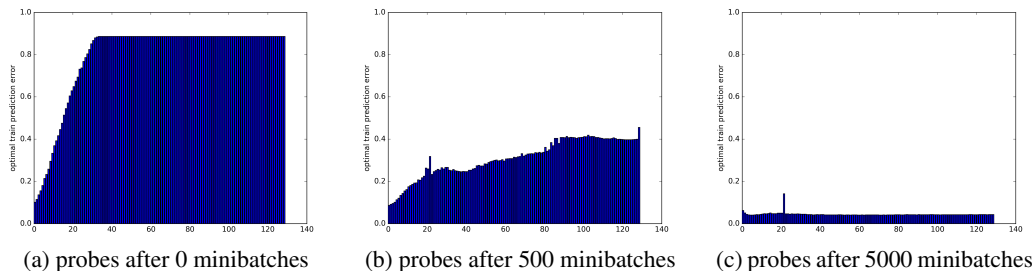

(a) probes after 0 minibatches  (b) probes after 500 minibatches  (c) probes after 5000 minibatches

Figure 7: A pathologically deep model with 128 layers gets an auxiliary loss added at every 16 layers (refer to simplified sketch in Figure 6a if needed). This loss is added to the usual model loss at the last layer. We fit a probe at every layer to see how well each layer would perform if its values were used as a linear classifier. We plot the train prediction error associated to all the probes, at three different steps. Before adding those auxiliary losses, the model could not successfully be trained through usual gradient descent methods, but with the addition of those intermediate losses, the model is "guided" to achieve certain partial objectives. This leads to a successful training of the complete model. The final prediction error is not impressive, but the model was not designed to achieve state-of-the-art performance.

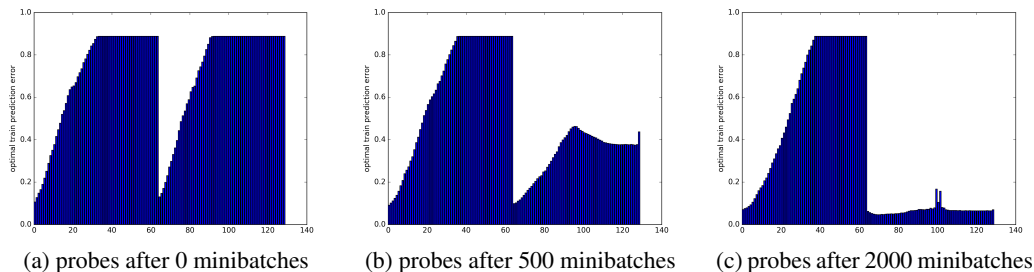

(a) probes after 0 minibatches  (b) probes after 500 minibatches  (c) probes after 2000 minibatches

Figure 8: A pathologically deep model with 128 layers gets a skip connection from layer 0 to layer 64 (refer to sketch in Figure 6b if needed). We fit a probe at every layer to see how well each layer would perform if its values were used as a linear classifier. We plot the train prediction error associated to all the probes, at three different steps. We can see how the model completely ignores layers 1-63, even when we train it for a long time. The use of probes allows us to diagnose that problem through visual inspection.

now we feel that it is premature to start using multi-layer probes. This also leads to the convoluted idea of having a regular probe inside a multi-layer probe.

## 5 CONCLUSION

In this paper we introduced the concept of the *linear classifier probe* as a conceptual tool to better understand the dynamics inside a neural network and the role played by the individual intermediate layers. We are now able to ask new questions and explore new areas. We have demonstrated how these probes can be used to identify certain problematic behaviors in models that might not be apparent when we traditionally have access to only the prediction loss and error.

We hope that the notions presented in this paper can contribute to the understanding of deep neural networks and guide the intuition of researchers that design them.

ACKNOWLEDGMENTS

Yoshua Bengio is a senior CIFAR Fellow. The authors would like to acknowledge the support of the following agencies for research funding and computing support: NSERC, FQRNT, Calcul Québec, Compute Canada, the Canada Research Chairs and CIFAR.

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

## A  APPENDIX

### A.1  PROPOSAL : TRAIN PROBES USING ONLY SUBSETS OF FEATURES

One of the challenges to train on the Inception v3 model is that many of the layers have more than $200,000$ features. This is even worse in the first convolution layers before the pooling operations, where we have around a million features. With 1000 output classes, a probe using $200,000$ features has a weight matrix taking almost 1GB of storage.

When using stochastic gradient descent, we require space to store the gradients, and if we use momentum this ends up taking three times the memory on the GPU. This is even worse for RMSProp. Normally this might be acceptable for a model of reasonable size, but this turns into almost 4GB overhead *per probe*.

We do not have to put a probe at every layer. We can also train probes independently. We can put probe parameters on the CPU instead of the GPU, if necessary. But when the act of training probes increases the complexity of the experiment beyond a certain point, the researcher might decide that they are not worth the trouble.

We propose the following solution : for a given probe, use a fixed random subset of features instead of the whole set of features.

With certain assumptions about the independence of the features and their shared role in predicting the correct class, we can make certain claims about how few features are actually required to assess the prediction error of a probe. We thank Yaroslav Bulatov for suggesting this approach.

We ran an experiment in which we used data $X \sim \mathcal{N}(0, I_D)$ where $D = 100,000$ is the number of features. We used $K = 1000$ classes and we generated the ground truth using a matrix $W$ of shape $(D, K)$. To obtain the class of a given $x$, we simply multiply $x^T W$ and take the argmax over the $K$ components of the result.

$$x \sim \mathcal{N}(0, I_D) \qquad y = \underset{k=1..K}{\arg\max} \left( x^T W[:, k] \right)$$

We selected a matrix $W$ by drawing all its individual coefficients from a univariate gaussian.

Instead of using $D = 100,000$ features, we used instead only $1000$ features picked at random. We trained a linear classifier on those features and, experimentally, it was relatively easy to achieve a 4% error rate on our first try. With all the features, we could achieve a 0% error rate, so 4 % might not look great. We have to keep in mind that we have $K = 1000$ classes so random guesses yield an error rate of 99.9%.

This can reduce the storage cost for a probe from 1GB down to 10MB. The former is hard to justify, and the latter is almost negligible.

## A.2    PROBES ON INCEPTION V3

We are interested in putting linear classifier probes in the popular Inception v3 model, training on the ImageNet dataset. We used the tensorflow implementation available online (`tensorflow/models/inception/inception`) and ran it on one GPU for 2 weeks.

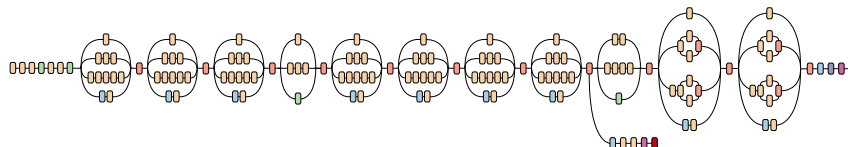

As described in section A.1, one of the challenges is that the number of features can be prohibitively large, and we have to consider taking only a subset of the features. In this particular experiment, we have had the most success by taking 1000 random features for each probe. This gives certain layers an unfair advantage if they start with 4000 features and we kept 1000, whereas in other cases the probe insertion point has $426,320$ features and we keep 1000. There was no simple "fair" solution. That being said, 13 out of the 17 probes have more than $100,000$ features, and 11 of those probes have more than $200,000$ features, so things were relatively comparable.

We put linear classifier probes at certain strategic layers. We represent this using boxes in the following Figure 9. The prediction error of the probe given by the last layer of each box is illustrated by coloring the box. Red is bad (high prediction error) and green/blue is good (low prediction error).

We would have liked to have a video to show the evolution of this during training, but this experiment had to be scaled back due to the large computational demands. We show here the prediction errors at three moments of training. These correspond roughly to the beginning of training, then after a few days, and finally after a week.

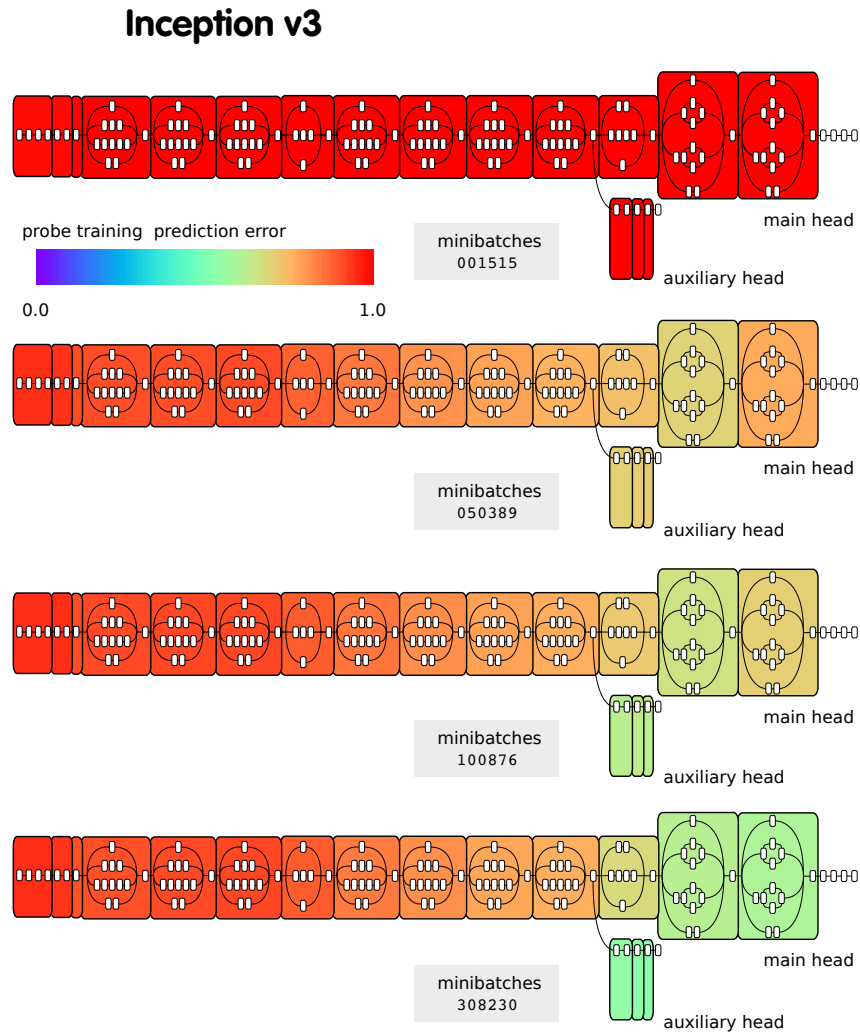

Figure 9: Inserting a probe at multiple moments during training the Inception v3 model on the ImageNet dataset. We represent here the prediction error evaluated at a random subset of 1000 features. As expected, at first all the probes have a 100% prediction error, but as training progresses we see that the model is getting better. Note that there are 1000 classes, so a prediction error of 50% is much better than a random guess. The auxiliary head, shown under the model, was observed to have a prediction error that was slightly better than the main head. This is not necessarily a condition that will hold at the end of training, but merely an observation.

