# Peer review of "Understanding intermediate layers using linear classifier probes"

_ICLR 2017 — rejected_

[Official Review · AnonReviewer3 · rating 4 · confidence 4 · 13 Dec 2016]
**Interesting problem, but technically and experimentally not solid enough**

This paper proposes to use a linear classifier as the probe for the informativeness of the hidden activations from different neural network layers. The training of the linear classifier does not affect the training of the neural network. 

The paper is well motivated for investigating how much useful information (or how good the representations are) for each layer. The observations in this paper agrees with existing insights, such as, 1) (Fig 5a) too many random layers are harmful. 2) (Fig 5b) training is helpful. 3) (Fig 7) lower layers converge faster than higher layer. 4) (Fig 8) too deep network is hard to train, and skip link can remedy this problem.

However, this paper has following problems:

1. It is not sufficiently justified why the linear classifier is a good probe. It is not crystal clear why good intermediate features need to show high linear classification accuracy. More theoretical analysis and/or intuition will be helpful.   
2. This paper does not provide much insight on how to design better networks based on the observations. Designing a better network is also the best way to justify the usefulness of the analysis.

Overall, this paper is tackling an interesting problem, but the technique (the linear classifier as the probe) is not novel and more importantly need to be better justified. Moreover, it is important to show how to design better neural networks using the observations in this paper.

[Official Review · AnonReviewer1 · rating 4 · confidence 4 · 13 Dec 2016]
**Linear predictiveness of intermediate layer activations.**

The authors propose a method to investigate the predictiveness of intermediate layer activations. To do so, they propose training linear classifiers and evaluate the error on the test set.

The paper is well motivated and aims to shed some light onto the progress of model training and hopes to provide insights into deep learning architecture design.

The two main reasons for why the authors decided to use linear probes seem to be:
- convexity
- The last layer in the network is (usually) linear

In the second to last paragraph of page 4 the authors point out that it could happen that the intermediate features are useless for a linear classifier. This is correct and what I consider the main flaw of the paper. I am missing any motivation as to the usefulness of the suggested analysis to architecture design. In fact, the example with the skip connection (Figure 8) seems to suggest that skip connections shouldn't be used. Doesn't that contradict the recent successes of ResNet?

While the results are interesting, they aren't particularly surprising and I am failing to see direct applicability to understanding deep models as the authors suggest.

[Official Review · AnonReviewer2 · rating 5 · confidence 3 · 18 Dec 2016]

This paper proposes a method that attempts to "understand" what is happening within a neural network by using linear classifier probes which are inserted at various levels of the network.

I think the idea is nice overall because it allows network designers to better understand the representational power of each layer in the network, but at the same time, this works feels a bit rushed. In particular, the fact that the authors did not provide any results in "real" networks, which are used to win competitions makes the results less strong, since researchers who want to created competitive network architectures don't have enough evidence from this work to decides whether they should use it or not.

Ideally, I would encourage the authors to consider continuing this line of research and show how to use the information given by these linear classifiers to construct better network architectures. 

Unfortunately, as is, I don't think we have enough novelty to justify accepting this work in the conference.

[Final Decision · Program Chairs · 06 Feb 2017]
**ICLR committee final decision**

The reviewers generally agreed that the research direction pursued in the paper is a valuable one, but all reviewers expressed strong reservations about the value of a linear probe on intermediate features. The lack of experiments on more complex state-of-the-art networks is also potentially problematic. In the end, it seems that there is insufficient evidence that the proposed approach is actually a useful tool in its present state, though the authors should be encouraged to pursue this line of research further.